Seroprevalence of SARS-CoV-2 anti-spike IgG antibody among COVID-19 vaccinated individuals residing in Surabaya, East Java, Indonesia

http://orcid.org/0000-0002-4452-5250 Megasari Ni Luh Ayu 1 2
Yamani Laura Navika 2 3
Juniastuti Juniastuti 2 4
Lusida Maria Inge 2 4
Mori Yasuko 5 ymori@med.kobe-u.ac.jp
1 Postgraduate School, Universitas Airlangga , Surabaya, East Java , Indonesia
2 Indonesia-Japan Collaborative Research Center for Emerging and Re-Emerging Infectious Diseases, Institute of Tropical Disease, Universitas Airlangga , Surabaya, East Java , Indonesia
3 Department of Epidemiology, Faculty of Public Health, Universitas Airlangga , Surabaya, East Java , Indonesia
4 Faculty of Medicine, Universitas Airlangga , Surabaya, East Java , Indonesia
5 Center for Infectious Diseases, Kobe University Graduate School of Medicine , Kobe, Hyogo , Japan
Mossong Joël
Electronic publication date: 2023 Sep 25
Publication date: 2023
Volume: 11
Electronic Location ID: e16142
Received 2023 Apr 6; Accepted 2023 Aug 29
Copyright: © 2023 Megasari et al.
Copyright year: 2023
Copyright holder: Megasari et al.
License: This is an open access article distributed under the terms of the Creative Commons Attribution License, which permits unrestricted use, distribution, reproduction and adaptation in any medium and for any purpose provided that it is properly attributed. For attribution, the original author(s), title, publication source (PeerJ) and either DOI or URL of the article must be cited.
License URL: https://creativecommons.org/licenses/by/4.0/

Keywords: SARS-CoV-2, COVID-19 vaccine, Anti-S-IgG, Indonesia, Seroprevalence

Funding: Japan Initiative for the Global Research Network on Infectious Diseases (J-GRID) Japan Agency for Medical Research and Development (AMED) Institute of Tropical Disease This work was supported by the the Japan Initiative for the Global Research Network on Infectious Diseases (J-GRID) from the Ministry of Education, Culture, Sport, Science and Technology in Japan, and the Japan Agency for Medical Research and Development (AMED); and the Institute of Tropical Disease as the Center of Excellence (COE) program by Kementerian Riset, Teknologi, dan Pendidikan Tinggi Republik Indonesia. The funders had no role in study design, data collection and analysis, decision to publish, or preparation of the manuscript.

==============================
Background

To limit the SARS-CoV-2 transmission, the Indonesian government launched a COVID-19 vaccination program in January 2021. Studies on the clinical treatment and implementation of COVID-19 vaccination have shown promising results; however, it is necessary to estimate the effectiveness of the vaccines. With the ongoing COVID-19 pandemic, studies have highlighted the impact of COVID-19 vaccines, especially CoronaVac, on Indonesian healthcare workers. To get a better picture of how the vaccines work in Indonesia, it is necessary to estimate the prevalence of SARS-CoV-2 anti-S IgG antibody induced by the COVID-19 vaccine in individuals who have already received two-to-three doses of vaccines.

Materials and Methods

Four-hundred and ninety-six whole-blood samples were collected from participants residing in Surabaya, East Java, Indonesia, who received a minimum of a two-dose COVID-19 vaccine. Serums were then isolated from the blood and subjected to detect SARS-CoV-2 anti-S IgG antibodies using a lateral flow immunochromatographic assay.

Results

The prevalence of positive anti-S-IgG antibodies was 91.7% (455/496) in all participants receiving a minimum of a two-dose COVID-19 vaccine. As many as 209 (85.3%) and 141 (96.6%) participants were seropositive for receiving CoronaVac and AstraZeneca, respectively. Meanwhile, all participants receiving two-dose CoronaVac with one booster dose of Moderna (105/100%) were seropositive (p < 0.05). Age, comorbidity, and time after the last vaccine were significantly correlated with seropositivity (p < 0.05).

Conclusion

Different vaccines might produce different antibody responses. Adopting a stronger policy regarding the administration of booster doses might be beneficial to elicit positive anti-S-IgG antibodies, especially among older individuals, those with comorbid diseases, and those with a longer time after the second vaccination dose.

Introduction

The World Health Organization (WHO) declared the novel coronavirus (COVID-19) outbreak as a global pandemic on March 11, 2020 (Cucinotta & Vanelli, 2020). COVID-19 came into place due to SARS-CoV-2, identified as the seventh member of the human coronavirus family (Song et al., 2022). Individuals with SARS-CoV-2 infection may experience a range of clinical manifestations, mainly grouped into asymptomatic, mild, moderate, severe, and critical infection (COVID-19 Treatment Guidelines Panel, 2022). The first positive COVID-19 case in Indonesia was detected on March 2, 2020 (Nugraha et al., 2020). Up to December 31, 2020, before the launch of the national COVID-19 vaccination program, a total of 743,198 people were confirmed cases, of which 22,138 were deaths with a case fatality rate of 3.0% (Manuhutu, 2021). A seroepidemiological study in 2020 reported that up to 13.1% individuals in Indonesia were previously infected by SARS-CoV-2 as confirmed by positive immunoglobulin (Ig)-G (IgG) antibodies (Megasari et al., 2021).

To limit the SARS-CoV-2 transmission, the Indonesian government implemented the COVID-19 vaccination program in January 2021. The government aimed to reach at least 70% of the vaccination coverage for 181.5 million people by March 2022 to achieve herd immunity. On March 5, 2023, the Indonesian Ministry of Health reported that 74.51% of Indonesian had received two full doses of vaccination, and 29.84% received an additional booster dose of COVID-19 vaccine (Indonesian Ministry of Health, 2023a).

The WHO recommended several COVID-19 vaccines to use. As of January 12, 2022, nine COVID-19 vaccines have been included in the Emergency Use Listing (EUL). These nine vaccines were the Pfizer/BioNTech Comirnaty (BNT162b2) (referred to as Pfizer), the SII/COVISHIELD and AstraZeneca/AZD1222 (ChAdOx1-S (recombinant)) (referred to as AstraZeneca), the Janssen/Ad26.COV2.S (referred to as Janssen), the Moderna (mRNA-1273) (referred to as Moderna), the Sinopharm COVID-19 (referred to as Sinopharm), the Sinovac-CoronaVac (referred to as CoronaVac), the Bharat Biotech BBV152 COVAXIN, the Covovax (NVX-CoV2373), and the Nuvaxovid (NVX-CoV2373) (World Health Organization, 2022). Six COVID-19 vaccines have been used nationwide for primary vaccines and booster in Indonesia. These six vaccines include CoronaVac, AstraZeneca, Pfizer, Moderna, Janssen, and Sinopharm (Indonesian Ministry of Health, 2022).

Vaccines are designed to induce virus-neutralizing antibodies. They prevent COVID-19 infection and avoid unfavorable clinical manifestations. Studies on the clinical treatment and implementation of COVID-19 vaccination have shown promising results on COVID-19 reduction. Reports on the phase III trials using a two-dose regimen showed 95% of Pfizer, 94.1% of Moderna, 67% of AstraZeneca, and 67.7% of CoronaVac efficiently reduced the virus transmission. Studies on individuals who were not previously infected by SARS-CoV-2 showed a two-dose COVID-19 vaccine was sufficient in preventing symptomatic infection (Polack et al., 2020; Baden et al., 2021; Olliaro, Torreele & Vaillant, 2021; Wei et al., 2021; Jin et al., 2022).

COVID-19 vaccines are designed to elicit an immune response against the spike (S) glycoprotein of SARS-CoV-2 required for SARS-CoV-2 binding, fusion, and cell entry. Vaccination induces the production of anti-S and anti-receptor binding domain (RBD) and neutralizes antibodies in the blood. Anti-S antibodies block the receptor-binding domain (RBD) on the S protein; thus, vaccines can prevent infection by inhibiting viral docking onto the angiotensin-converting enzyme 2 (ACE2) receptor. Despite producing several structural proteins, such as the S, nucleocapsid (N), membrane (M), and envelope (E), SARS-CoV-2 antibodies against N, M, and E have no neutralizing capability in the absence of anti-S antibodies. Similar to infection, vaccination results in early production of serum antibodies, including the SARS-CoV-2 anti-S-IgG and anti-S-IgM (Wisnewski, Campillo Luna & Redlich, 2021; Kyriakidis et al., 2021; Wang et al., 2021; Tan et al., 2022). Anti-S-IgG antibody was reported to exert neutralizing effects in vitro and become a useful indicator of an effective immune response, including the effectiveness of COVID-19 vaccines (Van Elslande et al., 2020; Rezaei et al., 2021). Detection and analysis of this parameter are relatively simple but provide a high correlation towards conventional neutralizing tests; thus, it can be highly utilized in both the COVID-19 pandemic and endemic (Lippi, Henry & Plebani, 2021; Jochum et al., 2022).

The various research settings may influence the outcomes of clinical trials. Continuous surveillance of vaccine effectiveness and the probability of waning immunity should be considered (Björk et al., 2022). In several African countries, the seroprevalence of SARS-CoV-2 anti-S antibodies in general population was lower than 90% (Awandu et al., 2022; Onifade et al., 2023), while in several Asian, European, and South American countries, the seroprevalence was reported to be more than 90% (Kajanova et al., 2022; Decarreaux et al., 2022; Aguilera et al., 2022; Elangovan et al., 2022). In Indonesia, several studies have highlighted the impact of COVID-19 vaccination on the elicitation of antibody responses. However, most studies are performed on healthcare workers receiving CoronaVac (Santi et al., 2021; Cucunawangsih et al., 2021, 2022; Hidayat et al., 2022). The seroprevalence of SARS-CoV-2-related antibodies among the general population, especially anti-S-IgG antibodies is rarely informed in Indonesia, while it plays an important role in neutralizing capability (Kyriakidis et al., 2021; Tan et al., 2022).

Up to December 30, 2020, East Java was reported having the second-highest cumulative SARS-CoV-2 infection among Indonesian Provinces, and Surabaya City had the highest cumulative cases in the province (Indonesian Ministry of Health, 2020; East Java Public Health Office, 2021). Thus, this study aimed to estimate the prevalence of SARS-CoV-2 anti-S IgG antibody of individuals of the general population, who received two to three doses of COVID-19 vaccines in Surabaya, East Java, Indonesia.

Materials and Methods

Ethical clearance

This study was approved by the Institutional Ethics Committee of Kobe University Graduate School of Medicine, Kobe, Japan (Approval No. B200600) and the Ethics and Law Committee of Airlangga University Hospital, Surabaya, Indonesia (Approval No. 163/KEP/2021).

Sample collection

Blood samples were collected in a medical research center in Surabaya, East Java, Indonesia, from September 2021 to March 2022. The inclusion criteria for participants determined include (1) Indonesian individuals who were aged 18 years or older; (2) had already received a minimum of a two-dose COVID-19 vaccine according to the Indonesia government’s guidelines and as proven by PeduliLindungi digital certificate (Indonesian Ministry of Health, 2021, 2023b); (3) were vaccinated at least 1 month before the research; and (4) formally consented to enroll in the study (proven by written informed consent letter). Exclusion criteria for the participants include (1) a history of COVID-19 vaccination dropout, (2) SARS-CoV-2 positive as diagnosed by real-time polymerase chain reaction (qPCR) or antigen test between the vaccination period and up to 6 months before sample collection, (3) the presence of COVID-19 related symptoms up to 1 month before sample collection, and (4) immune-system related disorder (immunodeficiency or auto-immune related diseases). Comorbidity was not included as an exclusion criterion. Peripheral blood samples and serum isolation were performed as previously described (Megasari et al., 2021).

Qualitative detection of SARS-CoV-2 anti-S IgG antibodies

SARS-CoV-2 anti-S IgG antibodies were qualitatively detected using a lateral flow immunochromatographic assay (Assure Tech, Hangzhou, China). The test was done by employing anti-human IgM antibody (test line IgM), anti-human IgG (test line IgG), and goat anti-mouse IgG (control line C) immobilized on a nitrocellulose strip. The conjugate pad contained recombinant SARS-CoV-2 antigen (antigen is recombinant nucleocapsid protein and spike protein (S1)) conjugated with colloid gold. The manufacturer’s reported values of the kit’s sensitivity and specificity in the COVID-19 IgG detection were 90% and 100%, respectively (Assure Tech, 2020).

Statistical analysis

Statistical analysis was performed using SPSS Statistics 17.0 (Advanced Analytics, Tokyo, Japan). A Chi-squared test of categorical variables was employed, and test results with p < 0.05 were considered significant.

Results

Demographic characteristics of research participants

This study collected 496 individuals who received a minimum of two-dose COVID-19 vaccine. Of the participants, 258 (52%) were male, and most were Javanese (464; 93.5%). Participants were classified into five age groups: 18–29, 30–39, 40–49, 50–59, and ≥60 years. Most participants were between 18 to 29 years of age (179; 36.1%). Based on employment, 108 individuals (21.8%) worked as medical or laboratory staff, while the rest (388; 78.2%) worked in non-medical sectors. Forty participants (8.1%) had one or more comorbid diseases, including diabetes mellitus, hypertension, dyslipidemia, and cardiovascular diseases. Most participants had taken treatment for 1 to 3 months after the last dose of the COVID-19 vaccine. All participants showed no COVID-19-related symptoms up to 1 month before sample collection, and reported no positive SARS-CoV-2 qRT-PCR or antigen test result up to 6 months before sample collection and between vaccination. Types of vaccine received were CoronaVac (245; 49%), AstraZeneca (146; 29.2%), and two doses of CoronaVac plus one booster (third) of Moderna (105; 21%). The demographic characteristics of participants are displayed in Table 1.

Table 1 Demographic characteristics and SARS-CoV-2 anti-S-IgG antibody seropositivity of study participants.

Characteristics	CoronaVac (n = 245)	AstraZeneca (n = 146)	Two doses of CoronaVac + 1 booster dose of Moderna (n = 105)	Total (n = 496)	
IgG positive n (%)	IgG negative n (%)	IgG positive n (%)	IgG negative n (%)	IgG positive n (%)	IgG negative n (%)	IgG positive n (%)	IgG negative n (%)	
Sex	p = 0.745	p = 0.755	N/A	p = 0.665	
Female	110 (84.6)	20 (15.4)	37 (97.4)	1 (2.6)	70 (100)	0 (0)	217 (91.2)	21 (8.8)	
Male	99 (86.1)	16 (13.9)	104 (96.4)	4 (3.6)	35 (100)	0 (0)	238 (92.2)	20 (7.8)	
Age (years)	p = 0.047*	p = 0.895	N/A	p = 0.000*	
18–29	93 (90.3)	10 (9.7)	42 (95.5)	2 (4.5)	32 (100)	0 (0)	167 (93.3)	12 (6.7)	
30–39	33 (84.6)	6 (15.4)	38 (97.4)	1 (2.6)	55 (100)	0 (0)	126 (94.7)	7 (5.3)	
40–49	32 (91.4)	3 (8.6)	42 (95.5)	2 (4.5)	11 (100)	0 (0)	85 (94.4)	5 (5.6)	
50–59	32 (78)	9 (22)	18 (100)	0 (0)	7 (100)	0 (0)	57 (86.4	9 (13.6)	
>=60	19 (70.4)	8 (29.6)	1 (100)	0 (0)	0 (100)	0 (0)	20 (71.4)	8 (28.6)	
Ethnicity	p = 0.003*	p = 0.496	N/A	p = 0.118	
Javanese	203 (86.8)	31 (13.2)	129 (96.3)	5 (3.7)	96	0 (0)	428 (92.2)	36 (7.3)	
Others	6 (54.5)	5 (45.5)	12 (100)	0 (0)	9	0 (0)	27 (84.4)	5 (15.6)	
Occupation	p = 0.136	N/A	N/A	p = 0.121	
Medical/laboratory personnel	14 (73.7)	5 (26.3)	0 (0)	0 (0)	89 (100)	0 (0)	103 (95.4)	5 (4.6)	
Non-medical/laboratory related job	195 (86.3)	31 (15.7)	141 (96.6)	5 (3.4)	16 (100)	0 (0)	352 (90.7)	36 (9.3)	
Time after the last dose of COVID-19 vaccine	p = 0.000*	p = 0.014*	N/A	p = 0.000*	
1 month	86 (96.6)	3 (3.4)	7 (77.8)	2 (22.2)	59	0 (0)	152 (96.8)	5 (3.2)	
2 months	47 (92.2)	4 (7.8)	24 (96)	1 (4)	46	0 (0)	117 (95.9)	5 (4.1)	
3 months	28 (80)	7 (20)	104 (98.1)	2 (1.9)	0 (0)	0 (0)	132 (93.6)	9 (6.4)	
4 months	8 (66.7)	4 (33.3)	6 (100)	0 (0)	0 (0)	0 (0)	14 (77.8)	4 (22.2)	
5 months	8 (61.5)	5 (38.5)	0 (0)	0 (0)	0 (0)	0 (0)	8 (61.5)	5 (38.5)	
≥6 months	32 (71.1)	13 (28.9)	0 (0)	0 (0)	0 (0)	0 (0)	32 (71.1)	13 (28.9)	
Comorbidity	p = 0.000*	p = 0.683	N/A	p = 0.000*	
Yes	14 (58.3)	10 (41.7)	6 (100)	0 (0)	10 (100)	0 (0)	30 (75)	10 (25)	
No	195 (88.2)	26 (11.8)	135 (96.4)	5 (3.6)	95 (100)	0 (0)	425 (93.2)	31 (6.8)	
Notes:

N/A, not applicable.

* significant (p < 0.05).

Prevalence of SARS-CoV-2 anti-S IgG antibody in vaccinated individuals

The prevalence of positive anti-S-IgG antibodies was 91.7% (455/496) in all two-dose vaccinated participants. Among recipients who received two-dose CoronaVac and AstraZeneca, 209 of 245 (85.3%) and 141 of 146 (96.6%) tested positive for anti-S-IgG antibody, respectively. All participants receiving two-dose of CoronaVac with one booster dose of Moderna (105/105; 100%) were anti-S-IgG positive. The difference in anti-S-IgG antibody seropositivity among individuals receiving a different type of vaccine was statistically significant (p < 0.05) (Fig. 1).

Figure 1 Seroprevalence of SARS-CoV-2 anti-S-IgG in study participants receiving different COVID-19 vaccines (n = 496).

Several characteristics were significantly correlated with the anti-S-IgG antibody seropositivity. Without considering the types of COVID-19, age, comorbidity, and time of vaccination after the last dose were significantly correlated to seropositivity (p < 0.05). The prevalence of seronegativity was higher among older individuals with comorbid diseases and a longer time after two primary or booster doses of vaccines. Based on the types of COVID-19 vaccine, a similar prevalence was also found in individuals receiving two-dose CoronaVac. In addition, individuals with an ethnicity other than Javanese had a higher prevalence of seronegativity than those of Javanese counterparts. The prevalence of anti-S-IgG antibody in vaccinated individuals can be seen in Table 1.

Discussion

The seroprevalence of SARS-CoV-2-specific antibodies varied worldwide. the administration of COVID-19 vaccines invoked the production of SARS-CoV-2 antibodies. In several African countries, the seroprevalence of anti-SARS-CoV-2 antibodies was found in different areas, such as 73.8% in Nigeria and 81.1% in Kenya (Awandu et al., 2022; Onifade et al., 2023). In other countries, such as India, Chile, French, and Slovakia, the seroprevalence was reported to be more than 90% (Ben Houmich et al., 2022; Kajanova et al., 2022; Decarreaux et al., 2022; Aguilera et al., 2022; Elangovan et al., 2022). Consistent with those findings, 91.7% of participants in this study exhibited anti-S-IgG antibody seropositivity.

This current study revealed that individuals receiving a two-dose primary vaccine i.e., CoronaVac and AstraZeneca had a significantly lower prevalence of anti-S-IgG antibody compared to those receiving two-dose CoronaVac with one booster dose of Moderna (p < 0.05). The seroprevalence of anti-S-IgG antibody was the lowest in CoronaVac recipients, followed by AstraZeneca and Moderna booster recipients.

Previous studies confirmed similar findings that anti-S-IgG antibody prevalence and titers of two-dose COVID-19 vaccine recipients were higher in mRNA-vaccine recipients, followed by AstraZeneca recipients, and the lowest were in CoronaVac recipients (Barin et al., 2022; Tan et al., 2022). Receiving two-dose CoronaVac and mRNA vaccine increased anti-S-IgG antibody titers, up to 7.9 times baseline titers a month after booster (Barin et al., 2022). Compared to a booster dose using CoronaVac, the mRNA vaccine produces higher anti-S-IgG antibody levels (Keskin et al., 2022; Barin et al., 2022). Thereby, it is recommended and safe to administer a heterologous booster vaccine, especially using mRNA vaccines, such as Moderna and Pfizer (Goh et al., 2022). Similar to this study, investigations on Indonesian healthcare workers receiving two-dose CoronaVac and one dose of Moderna booster also showed 100% anti-SARS-CoV-2 antibodies positivity (Cucunawangsih et al., 2022; Hidayat et al., 2022; Sinto et al., 2023).

Time after the last dose of vaccination significantly affected the anti-S-IgG antibody positivity, especially of CoronaVac recipients. A total of 3 months after vaccination, anti-S-IgG seronegativity turned to be around 20–38.5%, compared to 3.4% and 7.8% in the first and second months of the CoronaVac administration. A similar finding was observed in Malaysia, where individuals’ antibodies declined faster after receiving CoronaVac compared to AstraZeneca and mRNA vaccines. The rate of seropositivity was declining to 90% and 60% 3 months after the second vaccination of AstraZeneca and CoronaVac, respectively (Barin et al., 2022). Meanwhile, fourteen days after the second dose of CoronaVac, the level of SARS-CoV-2 anti-S antibodies was reported to decrease significantly by day 42, although persisting for up to day 98 (Cucunawangsih et al., 2021). Unlike individuals receiving only two doses of CoronaVac, those with an additional booster dose of Moderna showed 100% seropositivity up to 2 months after vaccination. Anti-SARS-CoV-2 antibodies were reported to decline in 90 days after the administration of either a homologous or heterologous booster (Lau et al., 2022).

However, an anomaly was observed among AstraZeneca recipients, particularly in the first month after the second dose. More than 20% of recipients (2/9; 22.2%) were seronegative, while the seroconversion rate in AstraZeneca recipients was reported to be more than 90% 2 weeks after the second dose (Varghese et al., 2022; Chau et al., 2022). This finding persisted likely due to the limited sample size or any unreported or undiagnosed participant’s condition, such as malignancies or immunosuppressive therapies, which might lower seroconversion to 70–85% (Thakkar et al., 2021).

Several participants’ characteristics were significantly correlated with anti-S-IgG antibody seropositivity, especially among CoronaVac recipients. Lower seropositivity was identified among older individuals, those with comorbid diseases, and those of non-Javanese ethnicity.

Older individuals, mostly more than 60 years old, experienced higher seronegativity after taking two doses of almost any type of COVID-19 vaccines, including CoronaVac, AstraZeneca, and Moderna (Fonseca et al., 2022; Chiarella et al., 2022; Malagón-Rojas et al., 2022). The seronegativity might happen due to immunescence, which may decrease the immune response due to the reduction of T-cell proliferation, naïve T-cell population, cytokine production, and antigen recognition (Fulop et al., 2009). Moreover, individuals with comorbid diseases, including diabetes mellitus, hypertension, or dyslipidemia, were more likely to be seronegative after receiving two-dose CoronaVac (Karamese & Tutuncu, 2022; Barin et al., 2022; Fonseca et al., 2022). Similar to age-related immunescence, comorbidities were reported to reduce vaccine immunogenicity and antibody response (Kwetkat & Heppner, 2020); thus, individuals with more comorbidities were likely to be seronegative with any types of COVID-19 vaccines received (Huang et al., 2023). An additional booster dose of the COVID-19 vaccine should be considered for older individuals and those with comorbid diseases (Fonseca et al., 2022). This recommendation lies in the current finding that all participants of all ages and those with or without comorbid diseases were anti-S-IgG antibody positive after receiving one booster dose of Moderna.

In addition to the influence of age and comorbid disease, all vaccines are supposed to elicit antibody responses in all individuals with different ethnicity and race. However, most studies reported the different efficacy of mRNA and recombinant vaccines among races, such as Caucasian, Asian, Hispanic, and African (Rahman et al., 2022; Salari et al., 2022). A study highlighting antibody responses among a specific ethnicity is very limited; thus, the current finding that mentions lower seropositivity in non-Javanese participants has no similar results to previous research. In general, no difference in seropositivity was observed among people with different ethnicity receiving either two-dose CoronaVac or AstraZeneca or among those receiving one booster dose.

This current study also did not find a significant difference in anti-S-IgG antibody seropositivity among individuals of different sex and occupation. However, previous studies described a higher risk of SARS-CoV-2 among healthcare workers despite vaccination (Shah et al., 2022); therefore, the booster is still needed to keep individuals, especially healthcare workers, safe. Previous studies found lower anti-S-IgG titers among males (Varghese et al., 2022; Barin et al., 2022; Fonseca et al., 2022). This current study also indicates a similar finding that participants of all sexes had a similar seropositivity level (Chau et al., 2022).

With the evidence generated from this study, it is important to note that this study still has several limitations. This study recruited a disproportionate sample size of participants receiving different types of vaccines (CoronaVac, AstraZeneca, and Moderna booster). Participants’ characteristics, including time after the last dose of vaccination, age, ethnicity, and comorbid disease, should also be considered when determining sample size. We recommend recruiting a proportionate sample size based on the inclusion criteria for future studies. Multicenter sampling might provide a better picture of COVID-19 vaccine seroprevalence as well. While exclusion criteria have addressed SARS-CoV-2 positive infection, not all participants received regular COVID-19 examination using either qRT-PCR or antigen test; thus, it was not exactly known whether all participants were SARS-CoV-2 negative up to 6 months before sample collection, between vaccination, or during sample collection. Despite six types of COVID-19 vaccines used in Indonesia, this study only managed to include three types of vaccines. Future studies are expected to address these limitations to better investigate the efficacy of COVID-19 vaccines.

Conclusions

The seroprevalence of positive anti-S-IgG antibodies was 91.7% among individuals residing in Surabaya, East Java, Indonesia, who received two to three doses of COVID-19 vaccines. Two-dose CoronaVac recipients exhibited the lowest seroprevalence compared to those receiving two-dose AstraZeneca and two-dose CoronaVac with one booster dose of Moderna. Higher seronegativity, especially among CoronaVac recipients, was correlated with a longer time after the administration of the second dose, older age, and comorbid disease. Administering a booster dose was beneficial to elicit positive anti-S-IgG antibodies; thus, this action can be used to strengthen booster vaccination policy, particularly in Surabaya and Indonesia in general.

Supplemental Information

Supplemental Information 1 Demographic characteristic, COVID-19 vaccination history, and anti-S-IgG antibody detection result of the study participants.

Click here for additional data file.

Additional Information and Declarations

Competing Interests

Author Contributions

Human Ethics

Data Availability

The authors declare that they have no competing interests.

Ni Luh Ayu Megasari performed the experiments, analyzed the data, prepared figures and/or tables, authored or reviewed drafts of the article, and approved the final draft.

Laura Navika Yamani performed the experiments, analyzed the data, prepared figures and/or tables, and approved the final draft.

Juniastuti Juniastuti performed the experiments, analyzed the data, prepared figures and/or tables, and approved the final draft.

Maria Inge Lusida conceived and designed the experiments, authored or reviewed drafts of the article, and approved the final draft.

Yasuko Mori conceived and designed the experiments, authored or reviewed drafts of the article, and approved the final draft.

The following information was supplied relating to ethical approvals (i.e., approving body and any reference numbers):

This study was approved by the Institutional Ethics Committee of Kobe University Graduate School of Medicine, Kobe, Japan (approval no. B200600) and the Ethics and Law Committee of Airlangga University Hospital, Surabaya, Indonesia (Ethical approval no. 163/KEP/2021)

The following information was supplied regarding data availability:

The raw data is available in the Supplemental File.

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
