# Peer review of "Seroprevalence of SARS-CoV-2 anti-spike IgG antibody among COVID-19 vaccinated individuals residing in Surabaya, East Java, Indonesia"

_PeerJ, doi:10.7717/peerj.16142_

## Round 0.1 · original submission · Major Revisions

Both reviewers agreed that your study needs some changes. They both raised issues regarding generisability to the wider Indonesian population due to the sampling design in a single facility. Please be more precise about the location of the study and that it is a single centre study in the abstract. Please address all of their concerns including limitations of the study design in the discussion before resubmitting.

Reviewer 1 ·

Basic reporting

This manuscript has clear, professional, relevant results and new data for Covid studies,

Experimental design

In the abstract, the authors stated 496 samples were collected, and in line 139 (result section) the authors stated “This study collected 479 individuals…” which one is correct?
 Reviewing the results of participants who are medical and non-medical personnel and also come from a certain region, these results cannot be generalized as Indonesian individuals. The title of this manuscript must be adjusted to the results of this research.
 Please explain in more detail the inclusion and exclusion criteria, such as the time of receiving the vaccine dose, the distance between vaccines, comorbidities, immune system disorders, participants who received the Pfizer vaccine?
 In the qualitative detection section, please (1) include the cut-off value of the anti-S IgG antibody used in this study; (2) add the reference for Assure Tech manufacturer procedure.
 Please add to the limitations of the study that the history of Covid-19 is a maximum of 1 month before sampling affects the prevalence/study results. Individual with a history of covid up to 6 months can increase SARS-CoV-2 anti-S IgG, how do the authors explain this related to sampling collection.
 The title in Figure 1 is more correct with "in study participants" instead of Indonesian. This is related to the sampling method used in this study.

Validity of the findings

No comment

Additional comments

There are several issues that need to be revised and clarified in this manuscript:.

Reviewer 2 ·

Basic reporting

Sligh improvement in English needed.

Experimental design

1. The urgency of the study to analyse the effectiveness of COVID-19 vaccine doses through the detection of antibodies against SARS-CoV-2 is not well elaborated. Authors should provide more reasonings to support the purpose of this study.

2. The introduction section also lacks information on the previous studies about detected antibodies against SARS-CoV-2 for the general population. To narrow down the purpose of the study, authors may want to explain whether there is inconclusive evidence in the literature regarding similar studies. The need for such a study to be conducted in a specific country, Indonesia, should also be addressed.

Validity of the findings

1. The choice of Surabaya as the case study and the reasonings for the selection of the medical centre should be elaborated in the method section (e.g., whether the COVID-19 cases were most prevalent in the city or the province of East Java).

2. Please be careful in claiming and concluding the results. Authors repeatedly use “Indonesians who received two to three doses of COVID-19 vaccines” in the study, while the fact is that the results are drawn from studying 479 observations coming from only 1 medical centre in a city in East Java. Generalising the results for the Indonesian population in the study may not be appropriate, and this should also be highlighted in the limitation section.

Additional comments

Authors may need to elaborate more on the reasonings behind the age and comorbidities as factors of seropositivity in the discussion section.

---

## Round 0.2 · accepted · Accept

All comments have been addressed.

Reviewer 1 ·

Basic reporting

No comment

Experimental design

No comment

Validity of the findings

No comment

Additional comments

None

Reviewer 2 ·

Basic reporting

No comment

Experimental design

No comment

Validity of the findings

No comment